# Lived experiences of disabled individuals living in Bahir Dar City, North West Ethiopia, a phenomenological study

Lijalem Jemberu[1], Yosef Wasihun[2], Tadele Fentabil Anagaw[2], Eyob Ketema Bogale[2] *

1 Tibebe Ghion Specialized Hospital, Bahir Dar, Ethiopia, 2 School of Medicine and Health Sciences, Department of Health Promotion and Behavioral Sciences, Bahir Dar University, Bahir Dar, Ethiopia

☯ These authors contributed equally to this work.
* ketema.eyob@gmail.com

## Abstract

### Background

When an individual's activities and performances in a normal environment are limited in nature, function, or quality, that person is considered to have a disability. Although many studies on disabled people's lived experiences have been conducted around the world, there is still a significant gap between nations in multiple cases such as culture, economic status, and the recommendation of a previous study in Ethiopia, which is one reason for conducting this research.

### Objectives

To explore the lived experiences of disabled individuals living in Bahir Dar City

### Methods

A descriptive phenomenology study design was employed in Bahir Dar city on 15 disabled individuals from November 15 to December 20, 2022. A heterogeneous purposive sampling technique was used to select study participants. Data was collected by using an in-depth interview. The rigor and trustworthiness of the study were maintained by transferability, dependability, credibility, and conformability. Colaizzi's phenomenological analysis method was used for the development of codes and themes. Software (ATLAS. ti 7) version 7.5.6 was used for analysis.

### Result

Five major themes and fourteen sub-themes were developed that explain lived experiences of disabled individuals. Physical, psychological, social, economic and coping strategy experiences were major themes. Depression and negative emotional behavior were sub-themes under psychological experiences. Unemployment with the absence of a workplace and inadequate income were sub-themes under the economical experiences of participants.

**Data Availability Statement:** All relevant data are within the paper and its Supporting Information files.

**Funding:** The author(s) received no specific funding for this work.

**Competing interests:** The authors have declared that no competing interests exist.

**Abbreviations:** AIDS, Acquired Immune Deficiency Syndrome; DNA, Deoxyribonucleic Acid; HIV, Human Immune Virus; ICF, International classification of functioning; PwDs, persons with disability UN: United Nation; USA, United States of America; WHO, World Health Organization.

## Conclusion

In this qualitative interview study, the lived experience of individuals living with disability in Bahir Dar city were addressed in terms of the physical, psychological, social, economic, and coping mechanism experience of disabled individuals. Special needs professionals and social support groups should have been assigned and present in all institutions to serve the PwDs to assure equal accessibility of services.

## Introduction

Disability is a condition of the body or mind or impairment that makes it more difficult for the person to do certain activities or activity limitation and interferes with participation restrictions [1, 2]. Disability is part of a human condition that can be experienced at some point in life either temporarily or permanently [3]. Currently, the World Health Organization International Classification of Functioning (WHO ICF) definition of disability is widely used, stating that a person has a disability if and when his or her activity performances in his or her usual environment are limited in nature, function, or quality of performance. It includes physical, learning, and intellectual disabilities that limit functional abilities. Vision, movement, thinking, remembering, learning, communicating, hearing, social relationships, and other disabilities [1, 4, 5].

Disability can be occurred since birth or through time after birth. It will happen due to an accident, war, or different diseases like cancer, heart attack stroke, or diabetes mellitus. Poverty is also one of the biggest causes of disabilities because people are forced to live and work in unsafe environments with poor needs access [3].

People with disabilities are more likely to have negative socioeconomic outcomes such as a lower level of education, poorer health outcomes, lower levels of employment, and higher poverty rates. Poverty may increase the risk of disability due to malnutrition, insufficient access to education and health care, hazardous working conditions, a polluted environment, and a lack of safe water and sanitation. Through a lack of employment and education opportunities, lower wages, and the increased cost of living with a disability, disability may also increase the risk of poverty [6].

According to UN statistics, there are currently over 600 million persons with disability worldwide, with 400 million living in developing countries and 80 million in Africa [7]. According to the World Health Organization (WHO) and the World Bank (WB), approximately 80% of the world's 1 billion people with disabilities (PwDs) live in developing countries where rehabilitation services are inadequate or non-existent. Population growth, man-made and natural disasters, war, accidents, and aging all contribute to an increase in these numbers, both globally and in developing countries [3, 8]. In Ethiopia, the prevalence rate is about 15% of the total population [3].

The effect of physical disability on people's lives is likely to be worse in developing countries than in developed economies because of the reliance on physical labor for income generation [9]. In many African countries, a range of beliefs, and attitudes underpin these alternative explanations. They include assumptions, misconceptions, traditional or religious beliefs and beliefs about the natural and supernatural worlds [10].

The need for having children is another challenge for persons with disability. For women's with disability who are in reproductive age the number of children with those need comparison is lower [1]. Other problems experienced by persons with disability are parents' negative attitudes towards the persons with disability, evidenced by hiding them [2].

Traditionally, Ethiopian society's perceptions of disability have stemmed from the religious and social backgrounds of the community. In most regions of the country, families with persons with disability children are considered to be punished as a consequence of the anger of the village witch doctor or an ancestral spirit. The community, without considering the impact on its members, displays humiliating and disabling attitudes toward people who have a disability [11]. The stigma associated with people with disabilities extended to people trying to help [12].

Although many studies on people with disabilities' lived experiences have been conducted around the world, there is still a significant gap between nations in areas such as culture and economic status. Especially when we come to our country, Ethiopia, we have a different culture from the rest of the world. There is no adequate previous study conducted on disability in this study area. Even the previous research entitled assessing hotel experiences of people with disability: in the case of Bahir Dar City, Ethiopia [13], recommends further study, and this study finding lack physical experience, economic experience, social experiences, psychological experience, and copying strategy. The purpose of this study was to gain an understanding of the essence of the experience of living with disability to address the aforementioned gap in the area.

## Methods and materials

### Study setting, design, and period

**Approach and setting.** A phenomenological study design was conducted in Bahir Dar City people who were living with a disability. Phenomenology study design helps to understand the meaning of people's lived experiences explores what people experienced and focuses on their experience of phenomena [14].

The study was conducted in Bahir dar city, which is the capital city of Amhara Regional State. Bahir Dar is the third biggest city in Ethiopia and it lies on the southern shore of Lake Tana, the country's biggest lake. As data from the office of Bahir Dar City Administration Women's, child and social service office in 2022 there are 2098 persons with disability permanently living in the city. The office categorized them into five types of disability. These are Movement restriction, blindness, deafness, mental developmental restriction, and leprosy. The study was conducted from November 15, 2022, to December 20, 2022.

### Population and sample

We recruited a total of 15 participants from Bahir Dar city from the date of November 15, 2022, to December 20, 2022. and data saturation was maintained. The participants were eligible for the study if they were living in Bahir Dar City for at least six months duration. Seriously ill patients who are unable to communicate at the time of the data collection period were excluded from the study.

Participants were selected from different places in the city. Then a written information sheet was provided for those selected participants who are persons with disability. After the people have been agreed, appointments were arranged in a place where it gives comfort, and informed consent was signed before starting the interviews.

A heterogeneous purposive sampling technique was used to select study participants. Heterogeneous purposive sampling help for looking to examine the diverse range of cases of disability that are relevant to the disability lived experiences phenomenon [15].

After having a supportive letter to the Bahir Dar city women's and social affairs office from the Bahir Dar city health directory, the office creates a link with city disability leaders and gave us a supportive letter to all sub-city social affairs and disability associations. Then we got all sub-city social affairs offices and disability associations and then we selected our study

participants from the community purposely. Sub-city disability association leaders were helpful in creating a connection with sub-city people with disability in their sub-city. The participants were included with various characteristics of different types of disabilities like blindness, movement restriction, leprosy, and deafness were involved. For deaf people a sign language translator was used.

## Data collection

An in-depth interview guide was used, following the phenomenological steps of bracketing, intuition, analyzing, and describing. This interview provided detailed information on the individual's experience, views, and feelings related to disability. The time taken for one interview ranged from 31 to 43 minutes. Data collection was taken over by the principal investigator and one note-taker.

The interviewer used open, non-directive questions that were adapted to the participant's common language. For deaf participants, a sign language translator was assigned to conduct the interview. The interview was initiated with a broad and general question followed by a probing question (interview guide as S2 File), and then the questions get more focused as the data collection progresses. During data collection, participants were engaged naturally, first asking the participant and listening attentively until they completed their idea, then probing based on the participant's response. In-depth interviews are conducted at a time and location that are convenient for the participants. Some interviews were conducted at their homes, others at their workplaces, and still others in empty spaces with limited sound and movement.

The interviews took place in a private setting in a safe environment to ensure the privacy of participants, which prevented unnecessary interruptions and increased their concentration. The interviews were also audio-recorded. After all, we had the phone numbers and addresses of all study participants in case the transcribed data needed to be verified.

## Ethics approval and consent to participate

Ethical approval for this study was obtained from the Institutional Review Board of Bahir Dar University (Approval number: CMHS/IRB/ 24/003/2022). All the participants were asked for their willingness to participate in the study before the interview and written consent was taken before the beginning. The IRB of Bahir Dar University allowed us to collect the data from people living with disability without collecting personal identifiers including names and they approved this consent procedure. The data from the participants (audio records and field notes) were kept in a proper way and codes were assigned to participants to assure confidentiality.

All the audio-recorded documents were deleted from the recording device after being copied to a password-protected personal computer. All the participants were aware of the right to withdraw and the right to not answer if they so wished.

**Data analysis.** All interviews and field notes were transcribed word for word in the Amharic language and then translated into English. The accuracy of the transcribed data was checked by listening and reading repeatedly every day after completing each interview, the field note taken was rewritten focusing on the nonverbal communication of the participant with incorporating the interview transcription.

After familiarization with the data code was assigned based on the context of the sentence line by line (code book as S1 File). After assigning code for each data with a similar meaning collected to the same category and form subthemes and then major themes were developed by merging themes those were having the same idea. Then a statement has been developed that

revealed complete a description of the lived experience of people with living in disabilities. Colaizzi's phenomenological analysis and (ATLAS. ti 7 software) were used.

**Rigor and trustworthiness of the study.** Trustworthiness was verified by addressing the dimension of credibility, transferability, dependability, and conformability.

*Transferability*. Interview was conducted with the aid of audio records. Then it was transcribed in Amharic language word by word after that it has been translated into English by the principal investigator and the transferability of this study was ensured by describing the thick description of study participants. Research context, setting, sample, and data collection procedure and assumptions clearly and in detail (in the next pages) that are central to the research to enable the reader to assess the findings' capability of being transferable. The strength and limitations of the study also have been described clearly.

*Dependability*. Dependability was increased by incorporating comments given by advisors and evaluators from the start of the proposal to the end of work in data analysis and discussion of the final categories, and obtained through deep communication with participants to develop trust in the interviewer.

*Credibility*. The credibility of the study was maintained by establishing a trusting and confidential relationship and prolonged engagement (we were called on the phone many times until our appointment for the interview was reached) to develop peer-like relationships with participants. The interview guide was evaluated by professionals before data collection. Also, to reduce the investigator's bias and the possibility of participant reactivity, the investigator's preconceptions were bracketed at the start of the study.

*Conformability*. Conformability of the study was ensured by providing rich quotes from the participants and detailed recording of each activity of the participant using a reflective journal in the time of the interview and every procedure of the study [16].

## Results

### Socio-demographic characteristics of respondents

A total of 15 in-depth interviews were conducted. Of those participants eight of them were single, six were married and one was widowed. Participants' types of disability were blindness, movement problem, deafness, and leprosy (Table 1).

### Themes and subthemes

Five themes and fourteen sub-themes emerged from this study. These themes are (1) physical experience with two sub-themes; poor infrastructure and individual physical condition, (2) economical experience with two sub-themes; unemployment with the absence of a workplace, and inadequate income. (3) social experiences with four sub-themes such as discrimination and stigma, disrespecting persons with disability, community misconceptions about disability, and unsatisfying institutional services; (4) psychological experiences with two sub-themes: negative emotional behavior and depression; (5) copying strategy with four sub-themes: self-esteem, family and friend support, entertainment, and religious practice (Fig 1).

### Theme 1: Physical experience

Under this major theme, there were two sub-themes found. These are poor infrastructure and people's physical conditions.

**Table 1. Socio-demographic information of study participants living in Bahir Dar City.**

| Participants | Age | Sex | Marital status | Educational level | Type of Disability | Years lived with disability |
|---|---|---|---|---|---|---|
| P1 | 28 | M | Single | 5<sup>th</sup> Grade | Movement | 16 |
| P2 | 27 | M | Single | Read and write | Movement | 15 |
| P3 | 27 | M | Single | 10<sup>th</sup> Grade | Movement | 27 |
| P4 | 31 | M | Married | 1<sup>st</sup> Degree | Blindness | 24 |
| P5 | 29 | M | Single | 11<sup>th</sup> Grade | Blindness | 16 |
| P6 | 23 | F | Single | 1<sup>st</sup> degree | Blindness | 8 |
| P7 | 27 | F | Married | 7<sup>th</sup> Grade | Blindness | 20 |
| P8 | 22 | F | Single | 7<sup>th</sup> Grade | Movement | 11 |
| P9 | 67 | M | Married | Non-literate | Leprosy | 49 |
| P10 | 28 | M | Single | 1<sup>st</sup> Degree | Blind | 21 |
| P11 | 28 | M | Married | Diploma | Deaf | 28 |
| P12 | 21 | F | Single | 12<sup>th</sup> Grade | Deaf | 18 |
| P13 | 29 | M | Married | Diploma | Deaf | 22 |
| P14 | 65 | M | Widowed | Non-literate | Leprosy | 47 |
| P15 | 43 | F | Married | 10<sup>th</sup> Grade | Leprosy | 24 |

## Subtheme 1: Poor infrastructure

Most participants listed that they have a lot of physical difficulty in their daily activities. These activities were compromised for various reasons, like poor building systems that don't include persons with disability conditions during construction, like the absence of elevators, no gates for wheelchairs, and the absence of handles at the steps of buildings. Participants informed us that cars are parked on the road due to ineffective traffic management systems. Sometimes different institutions, like electricity and telecommunication offices, may also leave the hole they dig after completing their job. Another issue raised by participants was the lack of modernized digital wheelchairs that would allow those to self-help.

*"There are many problems to getting daily bread and learning; for example, you will collide with poles and Bajaj's parked in the middle of the road, there are squares (KARETA) on the road, and even telecommunication may leave the hole it digs open when the job is finished."(29-year-old participant)*

## Subtheme 2: Individual physical condition

Some participants explained that difficulty integrating with the environment has affected their daily activities and made them dependent on their families and contracted vehicles for movement. Some others also explained that because of their physical condition, they weren't able to move and were forced to wait until help came from friends and family members. Other participants, because of their body discomfort while moving, are restricted from daily activity. Other participants reported feeling tired as a result of using artificial leg supports and an artificial leg.

*"As you look at me, my hands couldn't roll my wheelchair, and because of this, even to move from place to place, I would have to wait until my friends came to take me." "If my friends were unable to come, or unless I hired Bajaj drivers to take me home and carry me inside, I would have been forced to pass in the street overnight." (28-year-old participant)*

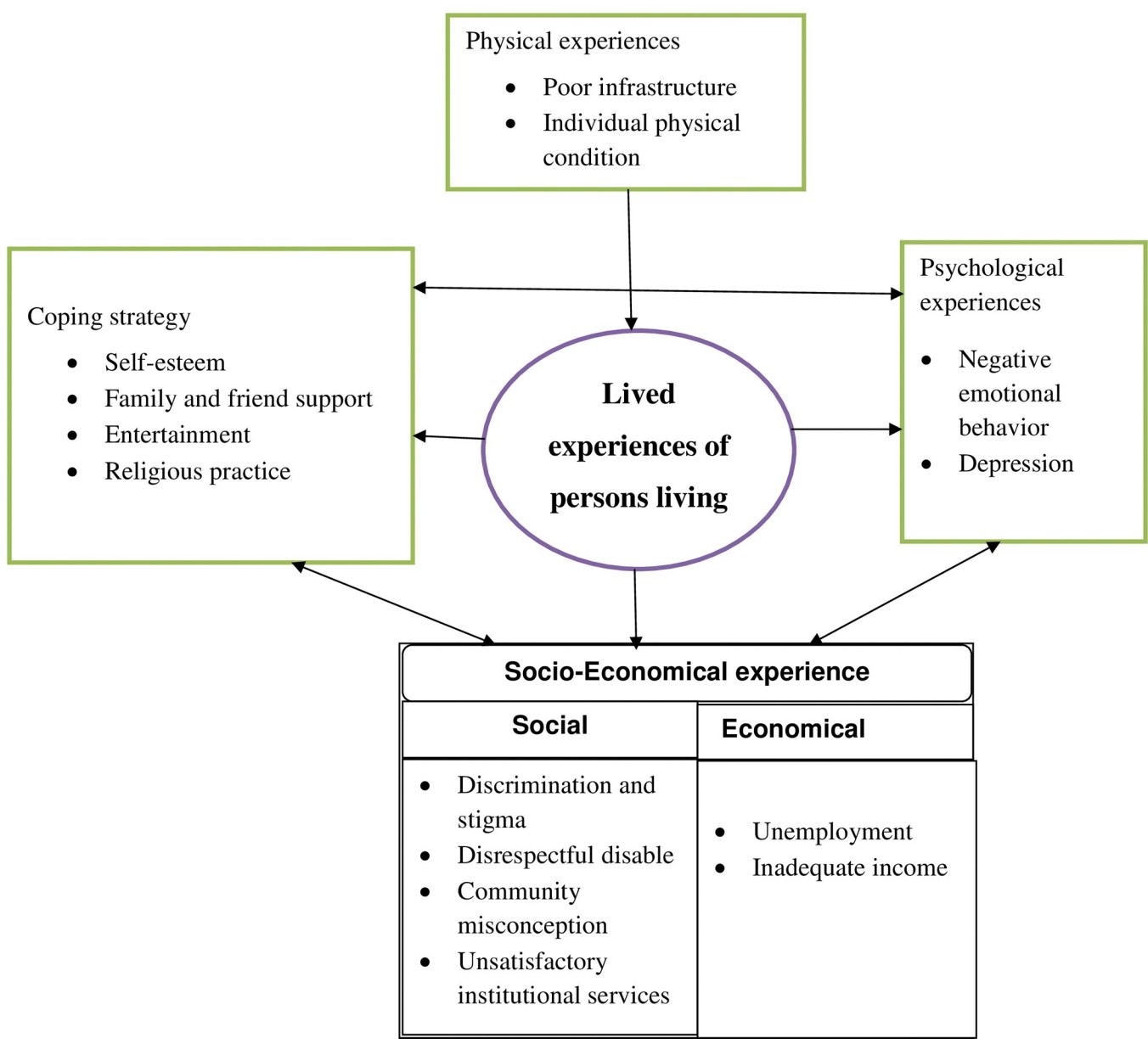

**Fig 1. Figural illustration of sub-themes and major themes.**

According to the sub-theme developed individual physical condition, participants explained that they are dependent on others for their daily activities; some others explained that they have a problem with self-care like washing their own body and clothes and daily activity unless friends and volunteer workers help them. Some of them also explained how it affects their education and their exam results. Another participant also told that he left the job he loves because of fatigue. A female participant with a leg brace recalled her school days in a rural area, a look of regret on her face, "at the time, I didn't think education would be this important," she explained.

*"When we go to school, my peers say hurry up, and when they get there, I'll be behind, then when I come back, I'll be behind, so I'm just going to get tired of it, so I left the school because of that." (23-year-old participant)*

### Theme 2: Psychological experience

Under this section. there were two sub-themes developed. These are negative emotional behavior and depression.

### Subtheme 1: Negative emotional behavior

Most of the participants explained that at the time of their disability occurrence, they had developed stressful conditions such as loneliness and irritability. Most of the participants explained that they were thinking so much about their disabilities that they thought they were the only persons with disability on earth at the beginning. Some other participants told that being irritable and naughty on every simple occasion was common. Findings in this study showed that sucking lips from the community side towards the persons with disability community creates a negative emotional behavior. Participants explained that they may be getting angry with a person who sucks their lip towards them, and this action is not sorrowful for the persons with the disability community. Especially for the blind community, the effect is high. According to the participants, it has a reminiscence effect on their disability and makes them feel as if they have lost something and do not act equally as other people.

*". . .in this city, a person approaches you for assistance and says, ohhhh [deep breath] (. . . .) it was better to lose your ear (. . .AYYYYY MINALE JOROHIN BAREGELIH NORO. . .) of course they said that for the matter of compassionate and kindness for us but it has negative implication in the blind community. Some blind people might insult and get angry with people who used to say that. . ." (28-year-old participant)*

*"Disability doesn't even have a proper name in rural areas. They used to say that Mr. X's daughter has been sick (YABA EGELIE LIJ BESHITEGNA HONALECH. . .). At that time, my behavior changed. "I become angry at small things because of my absence from school." (22-year-old participant)*

Another experience from one participant was phantom limb syndrome, which is a condition in which a person experiences the sensation of pain in a limb that does not exist. A male participant who lost his legs because of leprosy and lived with it said the following:

*"It was hard for me to forget losing my legs or accept it." One day when I woke up, I forgot my legs were amputated, so I tried to go downstairs without wearing my artificial legs to walk like normal people. What happened next was very sad. "I almost died [fell out of bed] (. . .)." (67-year-old participant)*

### Subtheme 2: Depression

Some of the participants said that they experienced feelings of hopelessness and sadness. Some participants explained that they did not expect to have a life like this because of their hopelessness. Some other participants also explained that they were losing interest in many things and even in life.

*"You may not be happy, like everyone else, but there is nothing you lose." They don't think we lived just like them. They used to say to us, "What will help you with things like this, and why are you learning where you will reach?" (27-year-old participant)*

On the other hand, some participants said that they enjoyed their life and are happy with their disabilities as other normal people do. And do not feel anything bad because of their disability. Some others also explained that they like to be feeling free and go to where many peoples are found to enjoy themselves.

*"My way of life is simple; I don't lead an extravagant lifestyle, and I laugh with those who laugh. If I have to drink, I drink; otherwise, that is my life philosophy. I am not lonely. I like playing and togetherness." (29-year-old participant)*

### Theme 3: Socio-economical experience

In this section on lived experiences, there are two major themes developed. These are social experiences and economical experiences.

### Subtheme 1: Economical experiences

In this section, there were two categories. These are (1) unemployment and absence of workplace and (2) inadequate income. Most participants have been persons with disability since childhood, and some others have been persons with disability since birth. As a result, the majority of them have no prior economic experience. However, some of them, those who became persons with a disability after experiencing economic experiences at the age of, have stated that there is a significant difference in economic experience before and after disability. One participant explained that he was the owner of a big house and a large farming area with a lot of castles. But he said everything was over, including his marriage when he became a person with a disability due to leprosy, and that he now spends his days begging.

### Category 1: Unemployment and lack of workplace

Some of the participants explained that they are suffering from economic problems as a result of unemployment, despite having graduated from university or having an interest in working as an employee, but they couldn't get the chance or were not assigned due to their disability. Some participants explained that because of unemployment, they are forced to work for a small salary as contract workers. Participants explained the reasons for unemployment as the current condition of the country [the war], COVID-19, and budget-related problems.

*"It is difficult to find a job because [government officials] give many reasons like the lack of budget, the present situation of the country [the war], the Coronavirus, and others related to attitude; it is difficult due to the laws and regulations, and they are challenges." (31-year-old participant)*

Some participants stated that the lack of a workplace and work stigma are the primary causes of their economic difficulties. Some participants with various types of work told us that they couldn't get a job because of the stigma associated with their field, which they said made life difficult with their family members.

*"For example, when I go to find woodwork, metalwork, and painting in my spare time, even though I can draw and work, look at this; all the drawings are mine [the wall drawing of the*

*classroom where the interview was conducted]. They have a bad face, and they are not willing to hire me." This is an attitude problem that is not being worked on. They believe that because I am deaf, I have no abilities. (28-year-old participant)*

Some other participants talked about workplace prohibitions by sub-city and kebele officials in different places. Some of them claimed that they were not permitted to sell shopping materials in bus stations. Some other participants complained about the prohibition of working on the street and begging. Also, one participant stated that providing a location for doing business is difficult because the area is located beyond the main road.

*"Even though the income I get is very little and wouldn't even cover the home rent price, I used to work weight-scale on the side of the street, but the kebele workers are making our life difficult." They forbid us from working here [on the street]. "My husband has 350 monthly payments from school as compensation for his blindness, so we are leading our lives this way." (27-year-old participant)*

## Category 2: Inadequate income

Some participants who are working on portable shopping told that the amount of profit they get from a single item is very small. Most participants explained that they have a problem fulfilling their basic needs. According to the findings of this study, some participants live on a low daily income, which does not allow them to eat properly, either in quantity or quality. Some participants also describe having a problem paying their rental home price.

*"I've been begging for money to keep my family together because my wife has been sick and has been in bed for the last 12 months, and my older daughter is also deaf ehhhh [deep breath]. . . After begging for some money, I bought an onion, a potato, and Shiro (an ingredient for Ethiopian traditional stew), then I gave it to my older daughter to prepare some food. I also purchased 5 injera (traditional Ethiopian food), which we will consume for lunch and dinner. (67-year-old participant)*

Some participants stated that they were unable to purchase an audio recording device for hearing after class sessions or other activities that required recording. Some other participants have problems fulfilling their physical assisting and moving devices, like accessories for their wheelchairs and accessories for their leg support, and have difficulty getting a rechargeable modern wheelchair to become self-reliant in their daily activities because of their inability to afford its price. One participant also claims that his disability could be treated but that he was unable to do so due to financial constraints.

## Subtheme 2: Social experiences

Some participants stated that before their disability, they were living harmoniously and happily with the community, but when their disability occurred, it became a loss, and after a long time and passing through various challenges, it had become as the previous. Some other participants also described that the social experience in the urban area is different from that in rural areas. Other participants, on the other hand, stated that there have been no social issues up to this present time.

## Category 1: Discrimination and stigma

The majority of participants in this sub-theme experienced family members' disability-related problems, such as not looking equally with other members of the family, drawing persons with

disability as special creatures that are not human beings, and the non-inclusiveness of the persons with the disability community in various programs. Some other participants said that they were locked inside their homes alone, not to be seen by other community members. Some participants also told that usually they are forced to fee at public toilets only because the home renters do not allow them inside their compound. Some others also told that they were forced to wash their body and cloth at Lake Tana because the home renters prohibit them to wash in their compound. A participant who lived with upper and lower extremity movement issues described how his parents make him miss school because they are afraid of the community.

*"When I asked my parents to take me to school, they responded, "Are you out of your mind? Don't you think what the community would say to us?" The community could tell us they send their child to school when they can't feed him. (. . .NEW LITASEDIBEN. . . ANTE TEMIREH MIN LITHON. . .) Then they kept me inside my home, not to be seen by other community members, for about 8 years." (28-year-old participant)*

Some participants have told that social activities like marriage are difficult for them; they are often discriminated against and stigmatized by the community because of the girls who want to marry them, and some girls will even change their minds because they are afraid of what the community will say about them and their parents.

*". . .she said she doesn't feel comfortable; her parents and brothers are insulting her; she said no, even her friends advised her not to do it; I have something to tell you, there is now a girl who married a person with disability person and lives away from her family." (27-year-old participant)*

However, one participant explained it in the opposite way that others did above. This blind participant explained that in all home decisions, he was the most accepted and listened to, and he was chosen to entertain when other people who were guests came to his home for a visit or different ceremonies, and his families were proud of him, and his mother took him to different ceremonies and celebrations. Another participant also explained that her family cares for her very much, and sometimes they would not like to allow her to go anywhere alone.

*". . . but my families were good to me despite the fact that I was the most influential person in our families, and they listened to me no matter what I said because they believed in me." "To your surprise, my mother did not volunteer to go to any celebration without me in case I refuse to go; she is also absent if I am absent from the program." (28-year-old participant)*

## Category 2: Disrespecting disabled people

A finding in this study shows that persons with disability have contempt for different types of social roles like marriage, Edir (the Ethiopian tradition of helping each other through association), and friendship. Other participants discussed the difficulties they faced while living in the community with their disability and being unable to be physically like others. Some participants also said that they have faced disrespectful practices from other parts of the community without their understanding.

*"When I say hello to greet a person, sometimes they say something to me—is it disrespectful or something?" (NIKET NEEW WEYIS MINDINEW YILUGNAL) Due to the fact that my hands are not normal (27-year-old participant)*

Some other participants stated that there is a challenge in their daily activities due to mis-communication with the community, such as difficulty on the road due to drivers who blow they can't listen, police officers who blow they may not listen, and even people they know who judge them based on their hearing problems. According to a deaf participant:

*"The police come to me and they think I have an ethical problem, so without any understanding, they usually slap me, which is very disgusting and disrespectful." It may be that he blew, but I didn't hear it. They slap me directly. It's very scary. Maybe all they said after this happened was an excuse; what will do for me after all? . . . (28-year-old participant)*

## Category 3: Community misconception about disability

Most of the participants explained that the community misunderstands them in different ways. They stated that most people do not believe they could live their lives as they do; most people believe that persons with disability are born from nothing or that their entire ancestors are persons with disability. Participants also stated that most people saw them as sinners and blamed their disability on their sin as a punishment from supernatural powers. Some others explain that the people thought they were persons with a disability because of Satan, the evil eye, or their parents' sin.

*"I haven't seen much isolation (laughing), but there are some things. For example, if I have a fight or disagreement with a normal person, even if the mistake is not mine, people say it's better to know your potential first because of your disability. They judged me, saying, "You are insane; how you could do that?" But it was not my fault, and this type of unfair judgment demoralizes me. (27-year-old participant)*

Some participants stated that the community fears touching them because it fears the wrong transmission; they also said some of the community may run away from them on the road while looking at them from the front. One participant also told how his mother left him when he was one year old, saying, "How could I have grown this type of baby?"

*What would I say, for example, if someone unexpectedly touched you on the road and assumed it would transmit to them? (EMIWARESBIGN YIMESLEWAL) One day I asked someone to help me pass the corridor. He came to me, and when he looked at my feet, he started to shake his hands. (28-year-old participant)*

## Category 4: Unsatisfactory institutional services

Most participants explained that accessing institutional services equally is a big problem for persons with disability. They said that getting some office managers to meet face-to-face is difficult since they do not allow it because they have poor reception towards persons with disability. In most public services, they have been told that they have served at last. There is also a communication gap in most institutions related to deaf people. According to the findings of this study, they are not receiving the services they are entitled to due to a lack of sign language translators.

*". . .When the farmer [who can't read and write] goes out to take something out, they cannot sign a special signature like us, but they give them money." So it is unfair to farmers, and we [blind people] are the same in terms of the situation where banks are told to be careful, but it's like farmers who cannot read and write are the same as we. They don't know whether they*

*have cheated or not. Why did the banks give it to us? They said it was for your sack, but I know that is a lie. (28-year-old participant)*

Another participant said the following about poor institutional services

*"Even when we went to the kebele, they didn't allow us to enter the head offices; they said it's not suitable for you [because it is on the 3$^{rd}$ floor]; but since my disability is a problem of sight, I could go and ask, but they are not volunteers." (27-year-old participant)*

### Theme 4: Coping strategy

Under the coping strategy, four major themes have been developed. Those are self-esteem, family and friend support, entertainment, and religious practice.

### Subtheme 1: Self-esteem

The majority of the participants had accepted their disability and thought it was nothing to worry about because of this they are confident in their selves and their self-esteem was good, and they need to be role models for the youth who are in different addictions at present. Participants explained that if they were not persons with disability, their lives might be in danger, and they thought they might not reach the position where they are at present. Others believed that being persons with a disability provided them with more opportunities to be disciplined than people their age. In terms of self-reliance, some participants stated that having a job makes them happy.

*"Sometimes I say it has happened to me for the better. My peers my age without disabilities are in serious trouble at present. "For example, they are smokers and chewers, and I am better than them in attitude, so I take my disability just because it happened to me for the better, and I don't think it could make me give up." (27-year-old participant)*

Whereas one participant explained that living with a disability means living with an incompetent way of life and defines disability as the inability to work and to live equally with other people.

*"Disability, to me, is the inability to work and support oneself." (67-year-old participant)*

### Subtheme 2: Family and friend support

For most of the participant's families and friends, support has a major role to run their life. Participants are hopeful that if anything happens to them, their families will make them comfortable. Participants explained that when they were feeling down, they considered moving somewhere unknown, but when they thought about their family, they changed their minds and chose to live with them again. Some other participants also reported that when something worries them, they enjoy finding their best friend and talking to them. Other participants explained they used to play with their friends, who could understand them, and with their kids to forget the different challenges they faced.

*"When I am worried about something, I find a place where there are deaf people, and I play where they are." I have fun with them. That means I am deaf, and they could communicate with me. However, other people must either know sign language or hear it for me to play or have fun with them. (21-year-old participant)*

Another participant said the following about his friend

*"Without him as my freshness for the city and coming from a rural area, I might lose myself and become demoralized and minor. I became a business owner because of him, he introduced me to many other people, and he is the reason I know so many things; I will never forget that day." (28-year-old participant)*

### Subtheme 3: Entertainment

The most common coping mechanisms used by participants in the sub-theme developed under entertainment are listening to FM radio, listening to music, watching movies, and finding friends to talk to. Respondents explained that after incidents of any angry feeling, they usually used to forget those occasions by doing activities like reading novels, watching football, and reading books.

*"I usually read books, listen to music, and talk to people, and it leaves stressful conditions and depression." "If I get a computer, I will not get tired of using it in my free time." (28-year-old participant)*

### Subtheme 4: Religious practice

To cope with their disability occurrence, most participants sought out persons with disability person who had a more severe disability type than they did. Participants are thanking God by comparing their disability type to others who have a severe disability type compared with them. Some other participants are also hopeful that, in the future, God may have a good plan for them. Some other participants said they used to cope by accepting what God gave them.

Most participants used to cope by listening to religious songs when stressful conditions happened to them. Participants told us that when they felt lonely and sad in the community because of their disability for various reasons, such as marriage-related stigma and discrimination, when they were undermined by others, or after a disagreement with someone, they used to pray to God to make them calm and think rationally.

*"At the time, I was very angry, and after a while, I prayed to God, please give me my patience and my mind to think the right thing." "At the time, I was very angry, and after a while, I prayed to God, please give me my patience and my mind to think the right thing." (28-year-old participant)*

## Discussion

The objective of this study was to explore the lived experience of persons with disability in Bahir Dar City. The findings of this study are organized with five themes and fourteen sub-themes. These themes are (1) physical experience with two sub-themes; poor infrastructure and individual physical condition, (2) economical experience with two sub-themes; unemployment with the absence of a workplace, and inadequate income. (3) social experiences with four sub-themes such as discrimination and stigma, disrespecting persons with disability, community misconceptions about disability, and unsatisfying institutional services; (4) psychological experiences with two sub-themes: negative emotional behavior and depression; (5) copying strategy with four sub-themes: self-esteem, family and friend support, entertainment, and religious practice.

This study finds out, buildings that are not designed with persons with disability in mind pose significant problems for persons with disability in terms of movement and daily activities.

This backs up a previous study on injuries conducted in Australia [17] which stated that walking up or down stairs without handles is difficult; even with handles, some stairs are difficult to use due to their steepness. This resemblance could be due to the fact that physical infrastructure faces the same challenges everywhere.

This study finds that participants were easily falling while using crutches, which is similar to a study conducted in [17], but an additional finding in this study was that they were afraid to walk with crutches and took a long time to adapt. This may be due to the difference in study areas and sociocultural differences between the two study areas.

In this study, the findings on blindness in terms of daily activity were that it was good to move from place to place except for one participant who only had restricted movement unless her mother helped, but another study shows that people with blindness were highly restricted from daily motion [18], this may be because of keeping them the same by restricting their movement, but in this study, as participants' conditions and explanations indicate, they have been forced to make a movement to meet their daily needs, and the presence of help during movement may be the reason to be active in their physical movement.

Of course, participants in this study had movement disabilities; one participant had completely restricted movement and was completely reliant on others for movement and self-care, including toileting and bathing, which supports a previous study done in Ethiopia [2]. The similarity in this finding could be due to the study area being in the same country, where socio-cultural similarities exist between the two study areas.

The majority of participants had adjusted their lifestyle so that conditions related to daily activities were simple for them, beginning with the job they chose. However, one study participant explained that he left a job he loved due to fatigue and that he now chose a job that was simple to do for his disability type. Otherwise, their mobility-related difficulties and difficulty in accomplishing their daily activities emerged because of the infrastructure in road and building plans. Previously Subjective experiences of difficulty were described as inaccessibility of facilities, fatigue, mobility-related difficulties, and difficulty in accomplishing activities of daily living were recognized [19], which was the same finding as this finding. This could be because physical experiences are objective.

We also learned from some of the participants in this study that they felt lonely and depressed, but the majority of the participants in this study were motivated to continue with their activity, despite facing various challenges. In the previous study, most participants lost interest in most of their activities and were usually depressed Gonder-Ethiopia [20] this could be a new finding difference associated with study area differences.

In this study, most of the participants have good self-esteem, even though they thought it happened for the better, and they accept what they have. Another study shows that people with blindness are highly lacking in their sense of self, contact with the real environment, and self-esteem [18]. This difference may be due to study area differences and the finding of a new result.

In this study, findings show that there is difficulty in fulfilling their needs, which is in line with the Study on Coping Strategies of Persons with disability in Residential Environments after Injury Events, where there was a finding that it was difficult to afford the price of a broken toilet seat and sometimes even though they had money, nobody could help them buy it [17]. But an additional finding in this study was that if they have money to buy something, they have the manpower to cooperate to do that. This difference may be due to the community differences between the two study areas.

In this study, participants explained that they are usually forced to eat food according to the day they have, not as they want, and they must because they can't afford to eat properly. Another participant told that she and her husband occasionally had difficulty paying the rental

price for the home where they live, which is nearly the same as the previous study, and the similarity of the study may be due to economic objectivity wherever it is.

This study finds out people's expectations and beliefs about the causes of disability: punishment from God by the sins of people and/or his/her parents, Satan, and the evil eye, which is the same as a previous study that supported causes of disability centered mainly on myths about disability caused by supernatural causes and disability caused by improper relationships by the parents in Kenya [12], but unlike those studies, luck from parents and belief in some objects like Idols and stopping their gift once they adapt were believed to cause disability. These additional findings may be due to the socio-cultural differences between the two study areas.

In this study, participants' self-esteem was good, and they believed that their disability happened to them for the better, and they thought if they were not persons with disability, their lives might not be like now, which they liked. Surprisingly, some of the study participants in this study believed that they should have to show the way for youths to become productive members of society and role models for people with disability community, even though they face different challenges. This is unlike other findings studied before [20]. This thought and attitude may stem from observing people with disabilities who are successful in their lives.

## Strengths and limitations

One of the strengths of this study is the sampling method. The method used for sampling is heterogeneous purposive sampling, used to capture the widest range of perspectives and possible findings with different backgrounds. Because of the difficulty of an in-depth interview with people who were grouped under the disability category, mental growth restriction people were not included because they couldn't explain their experiences.

## Conclusions

In this qualitative interview study, the lived experience of people living with disability in Bahir Dar city was addressed in terms of the physical, psychological, social, economic, and coping mechanism experience of persons with disability.

The physical experiences were poor infrastructure (absence of an elevator, absence of steep for a wheelchair at the building's gate and wrongly using roads) and personal physical conditions like movement problems, and self-care inability. Identified psychological experiences were negative emotional (like fear, sucking lips, anxiety, and anger) behavior and depression feeling lonely. The economical experiences were challenges related to unemployment (due to budget, covid-19 related, work stigma, and workplace problems) and low income. The social experiences were discrimination, stigmas, disrespecting, community misconception (thinking the cause of disability as it is God's punishment, sin, the anger of Idols, evil eye, Satan, and thinking disability is communicable by touch), and unsatisfactory institutional service.

Under the coping strategy, the identified experiences were self-confidence, self-reliance, friend support, friend support, entertainment (watching movies, reading books and nobles, football, FM radios), and religious practices like prying and listening to the religious song. Special needs professionals and social support groups should have been assigned and present in all institutions to serve the PwDs to assure equal accessibility of services.

## Supporting information

**S1 File. Code book.**
(DOCX)

**S2 File. Interview guide English version.**
(DOCX)

## Acknowledgments

We would like to give our special thanks to the study participants for their contributions to this study. Our deepest gratitude goes to Mrs. Wolela Endalew for her kind cooperation in sign language translation to conduct in-depth interviews with deaf participants.

## Author Contributions

**Conceptualization:** Lijalem Jemberu, Yosef Wasihun, Tadele Fentabil Anagaw, Eyob Ketema Bogale.

**Data curation:** Lijalem Jemberu, Yosef Wasihun, Tadele Fentabil Anagaw, Eyob Ketema Bogale.

**Formal analysis:** Lijalem Jemberu, Yosef Wasihun, Eyob Ketema Bogale.

**Investigation:** Lijalem Jemberu.

**Methodology:** Lijalem Jemberu, Yosef Wasihun, Tadele Fentabil Anagaw, Eyob Ketema Bogale.

**Project administration:** Lijalem Jemberu.

**Software:** Lijalem Jemberu, Yosef Wasihun, Tadele Fentabil Anagaw, Eyob Ketema Bogale.

**Supervision:** Yosef Wasihun, Tadele Fentabil Anagaw, Eyob Ketema Bogale.

**Validation:** Lijalem Jemberu, Yosef Wasihun, Eyob Ketema Bogale.

**Visualization:** Lijalem Jemberu.

**Writing – original draft:** Lijalem Jemberu, Yosef Wasihun, Tadele Fentabil Anagaw, Eyob Ketema Bogale.

**Writing – review & editing:** Lijalem Jemberu, Yosef Wasihun, Tadele Fentabil Anagaw, Eyob Ketema Bogale.

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
