## [Decision Letter · Decision Letter 0]

9 Mar 2023

PONE-D-23-03819Lived Experiences of Disabled Individuals Living in Bahir Dar City, North West Ethiopia, A Phenomenological studyPLOS ONE

Dear Dr. Bogale,

Thank you for submitting your manuscript to PLOS ONE. After careful consideration, we feel that it has merit but does not fully meet PLOS ONE’s publication criteria as it currently stands. Therefore, we invite you to submit a revised version of the manuscript that addresses the points raised during the review process.

We look forward to receiving your revised manuscript.

Kind regards,

Muhammad Arsyad Subu, Ph.D

Academic Editor

PLOS ONE

Journal Requirements:

2. Please describe in your methods section how capacity to provide consent was determined for the participants in this study. Please also state whether your ethics committee or IRB approved this consent procedure. If you did not assess capacity to consent please briefly outline why this was not necessary in this case.

Reviewers' comments:

Reviewer's Responses to Questions

**Comments to the Author**

1. Is the manuscript technically sound, and do the data support the conclusions?

Reviewer #1: Yes

2. Has the statistical analysis been performed appropriately and rigorously? 

Reviewer #1: Yes

3. Have the authors made all data underlying the findings in their manuscript fully available?

Reviewer #1: Yes

4. Is the manuscript presented in an intelligible fashion and written in standard English?

Reviewer #1: No

5. Review Comments to the Author

Reviewer #1: Reviewers comment

Date 6 March 2023

Title: Lived Experiences of Disabled Individuals Living in Bahir Dar City, North West Ethiopia, A Phenomenological study

Manuscript number: PONE-D-23-03819

Summary of the research and the overall impression

Title: A very impressive area of research that emphasized a special population group(disabled people) is a sound and good research area. A manuscript is also structured and written very well. Also, the article is presented in an intelligible fashion; however, the English language has to be edited further by a native English speaker.

Abstract: Exceeds 300-word counts and it may be beyond the journal requirement. No recommendations were made in the conclusion section of the abstract. I advise the author to include pertinent recommendations in the abstract section.

2. Discussion of specific areas for improvement

Minors issues

Introduction: The last paragraph says “Even the research done here in Bahir Dar recommends further study so that this study aimed to gain an understanding of the essence of the experience of living with disability”. Does it mean other research previously conducted on the same topic at Bahir Dar? If, what additional finding can the current study address to the scholarly community??

In the introduction or background section, the author needs to include a paragraph that shows problems related to the study phenomenon at Bahirdar, then he has to indicate the justification or aim for this study.

Method section

In the sentence “Then we got o all sub-city social affairs offices….” You have to remove “o” ? from the Population and sample heading in the fourth paragraph?

In the method section under scientific rigor, the author missed ensuring the reflexivity of the study. Reflexivity is essential in qualitative research because this field is heavily dependent upon the information that participants provide. Since questionnaires, discussions, and interviews are all led by researchers, the information gathered during qualitative studies may be influenced by underlying beliefs. So, make sure you’re actively reflecting on yourself and understand how your personal experiences impact every decision you make in your research. Do you have to explain how you maintained it?

In analysis section

Colaizzi's phenomenological analysis can be used reliably to understand people's experiences in descriptive phenomenology. Why did you use thematic over Colaizzi's phenomenological analysis for this research?

Result section

Sociodemographic table, I advise the author to remove the source of income column, it may expose the participants. It is better to anonymize the participant for ethical reasons?

Themes and subthemes

Under this section, tables of theme and sub-theme including a list of codes under each theme or codebook have to be uploaded as supporting information.

Discussion

This section could be started after indicating the brief objectives of the research followed by indicating the main finding of the study.

Under the conclusion section, this research has no recommendations. I advise the author needs to include the recommendation.

Generally, this paper is written very well and structured scientifically sound. It will add good pieces of evidence to the body of already available studies.

6. PLOS authors have the option to publish the peer review history of their article (what does this mean?). If published, this will include your full peer review and any attached files.

Reviewer #1: No

---

## [Author Response · Author response to Decision Letter 0]

16 Mar 2023

RESPONSE TO EDITOR AND REVIEWERS

RESPONSE TO EDITOR

RESPONSE: Thank you for coordinating the review process and fruitful comments. We have revised the manuscript and addressed Reviewer’s comments.

COMMENT: -1. Please ensure that your manuscript meets PLOS ONE's style requirements, including those for file naming. The PLOS ONE style templates can be found at  https://journals.plos.org/plosone/s/file?id=wjVg/PLOSOne_formatting_sample_main_body.pdf and https://journals.plos.org/plosone/s/file?id=ba62/PLOSOne_formatting_sample_title_authors_affiliations.pdf

RESPONSE: Thank you for your fruitful comments. We have ensured for you that our manuscript meets PLOS ONE's style requirements.

COMMENT:- 2. Please describe in your methods section how capacity to provide consent was determined for the participants in this study. Please also state whether your ethics committee or IRB approved this consent procedure. If you did not assess capacity to consent please briefly outline why this was not necessary in this case.

RESPONSE: Thank you for your fruitful comments. We have ensured for you that IRB of Bahir Dar university approved this consent procedure and we state how consent was determined during recruitment of the participant as:-

The IRB of Bahir Dar University allowed us to collect the data from people living with disability without collecting personal identifiers including name and they approved this consent procedure. 

COMMENT:- 3. In your Data Availability statement, you have not specified where the minimal data set underlying the results described in your manuscript can be found. PLOS defines a study's minimal data set as the underlying data used to reach the conclusions drawn in the manuscript and any additional data required to replicate the reported study findings in their entirety. All PLOS journals require that the minimal data set be made fully available. For more information about our data policy, please see http://journals.plos.org/plosone/s/data-availability.

Important: If there are ethical or legal restrictions to sharing your data publicly, please explain these restrictions in detail. Please see our guidelines for more information on what we consider unacceptable restrictions to publicly sharing data: http://journals.plos.org/plosone/s/data-availability#loc-unacceptable-data-access-restrictions. Note that it is not acceptable for the authors to be the sole named individuals responsible for ensuring data access. We will update your Data Availability statement to reflect the information you provide in your cover letter.

RESPONSE: Thank you for your fruitful comments. We have revised our data availability statement as:- All relevant data are within the manuscript and its Supporting Information files

COMMENT: -- 4. Please review your reference list to ensure that it is complete and correct. If you have cited papers that have been retracted, please include the rationale for doing so in the manuscript text, or remove these references and replace them with relevant current references. Any changes to the reference list should be mentioned in the rebuttal letter that accompanies your revised manuscript. If you need to cite a retracted article, indicate the article’s retracted status in the References list and also include a citation and full reference for the retraction notice.

RESPONSE: Thank you for your fruitful comments. We have reviewed our reference list and we ensured for you that they are complete and correct.

RESPONSE TO REVIEWER 1:

RESPONSE: We thank the reviewer for kind words and fruitful suggestion. We have addressed all comments as described below: -

Summary of the research and the overall impression

REVIEWER COMMENT: - Title: A very impressive area of research that emphasized a special population group(disabled people) is a sound and good research area. A manuscript is also structured and written very well. Also, the article is presented in an intelligible fashion; however, the English language has to be edited further by a native English speaker.

RESPONSE: We thank the reviewer for kind words and fruitful comments. We have revised it by editing English language errors. 

REVIEWER COMMENT: -Abstract: Exceeds 300-word counts and it may be beyond the journal requirement. No recommendations were made in the conclusion section of the abstract. I advise the author to include pertinent recommendations in the abstract section.

RESPONSE: We thank the reviewer for kind words and fruitful comments. We have revised our abstract section by fitting word count limit and by including pertinent recommendations in the abstract section.

2. Discussion of specific areas for improvement

Minors issues

REVIEWER COMMENT: -Introduction: The last paragraph says “Even the research done here in Bahir Dar recommends further study so that this study aimed to gain an understanding of the essence of the experience of living with disability”. Does it mean other research previously conducted on the same topic at Bahir Dar? If, what additional finding can the current study address to the scholarly community??

RESPONSE: We thank the reviewer for kind words and fruitful comments. We previous research is not on the same topic and we have revised it to increase the clarity of the research. 

REVIEWER COMMENT: -In the introduction or background section, the author needs to include a paragraph that shows problems related to the study phenomenon at Bahir Dar, then he has to indicate the justification or aim for this study.

RESPONSE: We thank the reviewer for kind words and fruitful comments. We have revised by showing problems related to the study phenomenon at Bahir Dar and , by indicating the j aim for the study.

Method section

REVIEWER COMMENT: - In the sentence “Then we got o all sub-city social affairs offices….” You have to remove “o” ? from the Population and sample heading in the fourth paragraph?

RESPONSE: We thank the reviewer for kind words and fruitful comments. We have removed “o” from the Population and sample heading in the fourth paragraph.

REVIEWER COMMENT: -In the method section under scientific rigor, the author missed ensuring the reflexivity of the study. Reflexivity is essential in qualitative research because this field is heavily dependent upon the information that participants provide. Since questionnaires, discussions, and interviews are all led by researchers, the information gathered during qualitative studies may be influenced by underlying beliefs. So, make sure you’re actively reflecting on yourself and understand how your personal experiences impact every decision you make in your research. Do you have to explain how you maintained it?

RESPONSE: We thank the reviewer for kind words and fruitful comments. We have revised it as Also, to reduce the investigator's bias and the possibility of participant reactivity, the investigator's preconceptions were bracketed at the start of the study.

REVIEWER COMMENT: -In analysis section, Colaizzi's phenomenological analysis can be used reliably to understand people's experiences in descriptive phenomenology. Why did you use thematic over Colaizzi's phenomenological analysis for this research?

RESPONSE: We thank the reviewer for kind words and fruitful comments. We have rwvised it based on your suggestion.

Result section

REVIEWER COMMENT: -Sociodemographic table, I advise the author to remove the source of income column, it may expose the participants. It is better to anonymize the participant for ethical reasons?

RESPONSE: We thank the reviewer for kind words and fruitful comments. We have removed source of income column from Sociodemographic table.

Themes and subthemes

REVIEWER COMMENT: -Under this section, tables of theme and sub-theme including a list of codes under each theme or codebook have to be uploaded as supporting information.

RESPONSE: We thank the reviewer for kind words and fruitful comments. We have uploaded codebook as supporting information.

Discussion

REVIEWER COMMENT: - This section could be started after indicating the brief objectives of the research followed by indicating the main finding of the study.

RESPONSE: We thank the reviewer for kind words. We have revised it by restating the objective of study and main finding of the study.

REVIEWER COMMENT: - Under the conclusion section, this research has no recommendations. I advise the author needs to include the recommendation.

RESPONSE: We thank the reviewer for kind words. We have revised our conclusion section by adding recommendation under the conclusion section.

REVIEWER COMMENT: - Generally, this paper is written very well and structured scientifically sound. It will add good pieces of evidence to the body of already available studies.

 RESPONSE: We thank the reviewer for kind words.

---

## [Editor Report · Decision Letter 1]

10 Apr 2023

L ived Experiences of Disabled Individuals Living in Bahir Dar City, North West Ethiopia, A Phenomenological study

PONE-D-23-03819R1

Dear Dr. Bogale,

We’re pleased to inform you that your manuscript has been judged scientifically suitable for publication and will be formally accepted for publication once it meets all outstanding technical requirements.

Kind regards,

Muhammad Arsyad Subu, Ph.D

Academic Editor

PLOS ONE
---

## [Editor Report · Acceptance letter]

9 May 2023

PONE-D-23-03819R1 

 Lived Experiences of Disabled Individuals Living in Bahir Dar City, North West Ethiopia, A Phenomenological study 

Dear Dr. Bogale:

I'm pleased to inform you that your manuscript has been deemed suitable for publication in PLOS ONE. Congratulations! Your manuscript is now with our production department. 

Kind regards, 

on behalf of

Dr. Muhammad Arsyad Subu 

Academic Editor

PLOS ONE